# Using Graphene-Based Composite Materials to Boost Anti-Corrosion and Infrared-Stealth Performance of Epoxy Coatings

**DOI:** 10.3390/nano11061603

**Published:** 2021-06-18

**Authors:** Meng-Jey Youh, Yu-Ren Huang, Cheng-Hsiung Peng, Ming-Hsien Lin, Ting-Yu Chen, Chun-Yu Chen, Yih-Ming Liu, Nen-Wen Pu, Bo-Yi Liu, Chen-Han Chou, Kai-Hsiang Hou, Ming-Der Ger

**Affiliations:** 1Department of Mechanical Engineering, Ming Chi University of Technology, Taishan, New Taipei City 243, Taiwan; mjyouh@mail.mcut.edu.tw; 2Department of Applied Science, R.O.C. Naval Academy, Zuoying, Kaohsiung 813, Taiwan; g981101@gmail.com; 3Department of Chemical and Materials Engineering, Minghsin University of Science and Technology, Xinfeng, Hsinchu 304, Taiwan; chpeng@must.edu.tw; 4Department of Chemical & Materials Engineering, Chung Cheng Institute of Technology, National Defense University, Dasi, Taoyuan 335, Taiwan; mslin479@gmail.com (M.-H.L.); g8184@yahoo.com.tw (T.-Y.C.); takululiu@gmail.com (Y.-M.L.); 5Department of Electrical Engineering, Yuan Ze University, Zhongli, Taoyuan 320, Taiwan; s1049001@mail.yzu.edu.tw (C.-Y.C.); s1070716@mail.yzu.edu.tw (B.-Y.L.); s1070720@mail.yzu.edu.tw (C.-H.C.); s1070714@mail.yzu.edu.tw (K.-H.H.); 6Chemical System Research Division, National Chung Shan Institute of Science and Technology, Longtan, Taoyuan 325, Taiwan

**Keywords:** graphene, anti-corrosion, epoxy, infrared stealth

## Abstract

Corrosion prevention and infrared (IR) stealth are conflicting goals. While graphene nanosheets (GN) provide an excellent physical barrier against corrosive agent diffusion, thus lowering the permeability of anti-corrosion coatings, they have the side-effect of decreasing IR stealth. In this work, the anti-corrosion properties of 100-μm-thick composite epoxy coatings with various concentrations (0.01–1 wt.%) of GN fillers thermally reduced at different temperatures (300 °C, 700 °C, 1100 °C) are first compared. The performance was characterized by potentiodynamic polarization scanning, electrochemical impedance spectroscopy, water contact angle and salt spray tests. The corrosion resistance for coatings was found to be optimum at a very low filler concentration (0.05 wt.%). The corrosion current density was 4.57 × 10^−11^ A/cm^2^ for GN reduced at 1100 °C, showing no degradation after 500 h of salt-spray testing: a significant improvement over the anti-corrosion behavior of epoxy coatings. Further, to suppress the high IR thermal signature of GN and epoxy, Al was added to the optimized composite at different concentrations. The increased IR emissivity due to GN was not only eliminated but was in fact reduced relative to the pure epoxy. These optimized coatings of Al-GN-epoxy not only exhibited greatly reduced IR emissivity but also showed no sign of corrosion after 500 h of salt spray test.

## 1. Introduction

Corrosion is a universal problem in daily life. How to protect metals against corrosion has always been an important topic in all industry fields. Many corrosion prevention technologies have been developed and applied to ships, airplanes, automobiles, industrial piping, buildings, etc. Among them, organic coatings are the most widely used because of their low cost and good corrosion resistance in harsh corrosive conditions. However, pure organic coatings cannot block moisture and other corrosive agents in the environment (chloride ions, oxygen, sulfides, etc.) for a long time. Therefore, in recent years, many researchers have added nanomaterial fillers such as silica, nanoclay, graphene, etc. into organic coatings [1,2,3,4,5,6,7,8,9,10,11,12] to extend the diffusion paths of corrosion agents and thus improve the corrosion resistance.

Graphene is a two-dimensional material that has great potential as an anticorrosion filler because of its sheet-like geometry and high aspect ratio, which are favorable for the physical barrier effect. Furthermore, its hydrophobicity and exceptional impermeability [13,14,15,16] to all gases and liquids are key elements for corrosion resistance. Consequently, graphene-based anticorrosive coatings have attracted great attention in recent years. For example, several research teams [17,18,19,20,21] have independently confirmed that graphene prepared by chemical vapor deposition (CVD) can effectively inhibit the oxidation of copper and nickel-copper alloys. However, defects in, or scratches on, these ultrathin graphene films may easily deteriorate their function. For long-term protection, thick organic coatings with added graphene fillers are more practical. For example, Yu et al. [22] added titanium dioxide/graphene into an epoxy resin to form a composite coating. They reported that graphene flakes improved the corrosion resistance by blocking the micro-pore corrosion channels formed in epoxy due to solvent evaporation. Chang et al. [6] demonstrated that 0.5 wt.% of graphene filler incorporated into polyaniline matrix significantly raised the corrosion protection efficiency (from 4.82% to 53.49%), while the same amount of clay filler offered only 12.43% efficiency. They proposed that sheet-like fillers, with a much higher aspect ratio than clay, can provide better physical barrier effect by forcing the corrosive agents to take a more torturous path through the coating. The optimized graphene loading varied from ~0.1 wt.% to ~8 wt.%, depending on the choice of matrix material, the dispersion uniformity of filler and the preparation/modification of graphene [6,7,8,9,10,11,12,22,23,24,25,26,27,28].

In some applications, such as architecture, thermal protection and infrared stealth, low infrared (IR) emissivity coatings are demanded (e.g., to reduce the IR signature of a military target to conceal it from IR detection). It would be very useful to incorporate IR camouflage into the anticorrosion coatings. However, adding graphene in the coating tends to increase the IR emissivity because graphene has a wide absorption band in IR. Metal powders such as Al, Zn, Cu and Ni are often used as low-IR-emissivity pigments [29,30,31] for their high reflectivity, hence small absorption and low emissivity. Al powders are very frequently used owing to their low cost and low density [29,30,31,32,33]. In this work, graphene and Al powders are added to epoxy resin coatings to perform the functions of anticorrosion and low-emissivity pigment, respectively. The result is simultaneous high corrosion resistance and good IR camouflage.

## 2. Experimental Details

### 2.1. Graphene Nanosheet (GN) Fabrication

Natural graphite (200 mesh) obtained from Alfa Aesar (Heysham, Lancashire, UK) was first oxidized into graphene oxide (GO) using the Staudenmaier method [34,35,36]. First, sulfuric acid (87.5 mL) and nitric acid (46 mL) were mixed well by stirring 15 min in an ice bath, and then graphite powder (0.5 g) was dispersed into the solution. After 15 min, potassium chlorate (5.5 g) was added into the system--very slowly to prevent strong reaction. After reacting for 96 h, the mixture was diluted by de-ionized (DI) water and then filtered. The GO was repeatedly rinsed in a 5% solution of HCl for three times. It was then washed continually with DI water until the pH became neutral. The GO slurry was dried and pulverized. Finally, 1 g of this GO was heated to 1100 °C in Ar atmosphere and held in the furnace for 1 min to be reduced/exfoliated into few-layer graphene nanosheets (GN), or reduced graphene oxide (rGO). To investigate the effects of the residual oxygen function groups on GN’s anticorrosion performance, we also prepared GN samples at lower reduction temperatures (300 °C and 700 °C) for comparison.

### 2.2. Preparation of Composite Coatings

Composite coatings were prepared from various amounts of GN (0.01, 0.05, 0.1 and 0.5 wt.%) and Al powder (APS 11 micron, 99.7%, CAS:7429-90-5, Echo Chemical Co., Toufen, Taiwan) at 0, 5, 15, 25 and 35 wt.%, which were added into the epoxy resin (base (Intergard 263, AkzoNobel):curative (FAA262, AkzoNobel) = 4:1). To prepare a well-dispersed composite, the mixture was first pre-mixed with a planetary centrifugal mixer (Thinky, AR-250, Tokyo, Japan) at 2500 rpm for 10 min. After pre-mixing, the mixture was then fed into a three-roll mill (Exakt, 80E, Oklahoma City, OK, USA) repetitively for 9 times in order to improve the dispersion of graphene. The first and third rollers rotated in the same direction while the center roller rotated in the opposite direction. The angular speed ratio of the three rollers was 1:3:9—this speed difference created high shear forces, which were necessary to break up groups of agglomerates. The widths of the gaps between the rollers were reduced gradually from 100 μm to 50 μm to 30 μm in order to improve the dispersion uniformity. Subsequently, the mixture was degassed by the planetary mixer at 1000 rpm for 5 min. The resulting mixtures were applied on substrates (mild steel panels with a dimension of 6.0 cm × 6.5 cm × 1 mm) using a spray gun set at a pressure of 35 psi. A non-destructive coating thickness gauge (Automation Dr. Nix, QNix^®^ 4500P, Cologne, Germany) was utilized to measure the thickness of the dried coating. The coating thickness at the center and four corners was measured to confirm that the coating was uniform in thickness. The dry film thickness was controlled to be 100 ± 10 μm.

### 2.3. Characterization

The microstructures of the graphite, GO and GN samples were characterized by X-ray diffractometry (XRD, Rigaku, D/Max 2200, Tokyo, Japan). The surface morphology of GN, Al powder and composite coating samples were examined with scanning electron microscopy (SEM, Hitachi, S-3000N, Tokyo, Japan). The Brunauer–Emmett–Teller (BET) specific surface area for GN was determined from the N_2_ physisorption isotherms obtained with a Micromeritics (Norcross, GA, USA) ASAP 2020 surface area and porosity analyzer. The Raman spectra of the GN samples were obtained with a Renishaw (Wotton-under-Edge, UK) inVia^TM^ Raman microscope using a 514.5-nm laser excitation wavelength. The elemental compositions of the GO and GN samples were determined by elemental analysis (EA, Thermo Scientific, Flash 2000 CHNS-O Analyzer, Waltham, MA, USA). Polarization tests on coated mild steel specimens were carried out with an Metrohm Autolab (Utrecht, the Netherlands) PGSTAT 302N potentiostat equipped with a frequency response analyzer. Ag/AgCl was used as the reference electrode, and Pt was used as the counter electrode. Polarization-curve measurements were performed in a 3.5 wt.% NaCl solution at room temperature. The electrochemical impedance spectroscopy (EIS) measurements for all coated samples immersed in 3.5 wt.% NaCl solution were performed with the same potentiostat within a frequency range from 100 kHz to 10 mHz. The relative corrosion resistance of the coated mild steel specimens was further verified with the salt spray test (SST) in accordance with ASTM (American Society for Testing and Materials) B117. To measure the IR emissivity, IR images in the windows of 3–5 and 8–12 μm were first acquired using Teledyne FLIR (Thousand Oaks, CA, USA) ThermaCAM Merlin and SC2000 infrared cameras, respectively. The IR emissivity was then obtained following ASTM E1933-99a (standard test methods for measuring and compensating for emissivity using infrared imaging radiometers).

## 3. Results and Discussion

### 3.1. Corrosion Resistance of GN-Filled Composite Coatings

The XRD patterns for the natural graphite powder, Staudenmaier GO and GN reduced at 1100 °C, are displayed in Figure 1. The (002) peak of graphite shifts significantly from 26.3° to 12.3° after oxidation with acid treatment. This indicates a much larger interlayer spacing due to the intercalation of acid. The successful exfoliation of graphene sheets can be verified by the vanishing of (002) peak in GN after the thermal reduction step. The BET specific surface area for GN calculated from nitrogen adsorption isotherm (not shown) was 517 m^2^/g, which also confirmed that a high degree of exfoliation was achieved. As in many applications of graphene in composite materials [35,36,37,38,39,40], a large specific surface area for the graphene filler and a highly uniform dispersion in the matrix are essential to ensure the optimal performance.

To find the best filler content for our anticorrosion coating, samples of epoxy with 0, 0.01, 0.05, 0.1 and 0.5 wt.% of GN (reduced at 1100 °C) were prepared and tested. Figure 2 shows the potentiodynamic polarization curves for epoxy coatings added with various GN contents. Table 1 lists the electrochemical parameters: corrosion potential (*E_corr_*), corrosion current density (*i_corr_*) and corrosion rate (*v_corr_*), which is defined as [10,28]:$$v_{corr}=\frac{87600E_{w}i_{corr}}{n\rho F}(\mathrm{mm}/\mathrm{year})$$
where *E_w_* is the formula weight of carbon steel (55.85 g/mol), *ρ* is the density of carbon steel (7.85 g/cm^3^), *n* is the chemical valence of the ions and *F* is the Faraday constant (96,485 C/mol = 26.80 A h/mol). As seen from the polarization curves, the pure epoxy coating shows the lowest *E_corr_* of -0.48 V. All GN-filled composite coatings exhibit more positive *E_corr_*, and the values increase monotonously with increasing content of GN. This increase in *E_corr_* is due to the change of surface composition in the coating and indicates that the corrosion tendency is suppressed by introducing the GN filler. In terms of corrosion current, as the GN content increases, *i_corr_* drops first (for GN content from 0 to 0.05 wt.%) and then increases at higher GN contents (0.1 and 0.5 wt.%). However, all GN-filled coatings show significantly lower *i_corr_* and *v_corr_* than the pure epoxy coating, indicating that the GN fillers in the coatings effectively suppress the cathodic and anodic reactions on the mild steel substrate, hence a reduced corrosion rate. According to the Butler–Volmer equation for a reaction in which the rate is limited by the activation overpotential *η* (i.e., *E* − *E_corr_*),
$$i=i_{corr}\left[\exp\left(\frac{2.303\eta }{\beta_{a}}\right)-\exp\left(\frac{-2.303\eta }{\left|\beta_{c}\right|}\right)\right]$$
where *β_a_* (V/dec) and *β_c_* (V/dec) are the Tafel slopes of the anodic and cathodic branches of the potentiodynamic polarization curve, respectively. The values of *β_a_* and *β_c_* for all samples are also listed in Table 1. These slopes represent the overpotential required to increase the cathodic or anodic reaction rates by a factor of 10. The GN_0.05_/epoxy coating exhibits the lowest corrosion current density (4.57 × 10^−11^ A/cm^2^) and the slowest corrosion (5.31 × 10^−7^ mm/year), which are more than 3 orders of magnitude lower than the pure epoxy coating, indicating a significant improvement in corrosion resistance due to the physical barrier effect of GN flakes. However, when the GN content increases above 0.05 wt.%, *i_corr_* starts to increase rather than keeps decreasing, in contrast to the monotonous increase in *E_corr_*. Note that *i_corr_* is a kinetic value and *E_corr_* is a thermodynamic property: the corrosion rate is proportional to *i_corr_*, while *E_corr_* only shows corrosion tendency (but not the corrosion rate). The corrosion current is strongly dependent on the integrity of the coating, which is related to the number and sizes of voids, pores and cracks within the coating. The higher *i_corr_* values at higher GN contents can be attributed to the increasing difficulty of dispersing nanomaterials uniformly in the composite, which results in more voids and defects within the coating.

For the pure epoxy coating, the anodic polarization branch near *E_corr_* initially shows a narrow region of low current density (on the order of 1 × 10^−7^ A/cm^2^), but starting from +130 mV vs. *E_corr_* (i.e., −350 mV) the current density rises steeply by two orders of magnitude (from ~1 × 10^−7^ A/cm^2^ to ~1 × 10^−5^ A/cm^2^), indicating accelerated dissolution of the anode (probably due to the onset of a different anodic reaction, localized corrosion or other corrosion mechanisms) at a higher overpotential. In contrast, a markedly different shape of polarization curve is found for all GN-filled composite coatings: a wide “passive” region with a slow anodic dissolution is observed. The low anodic “passive” current density, which results from the very effective suppression of corrosion kinetics at the anode, indicates that these GN-fillers in the coatings provide significantly improved protection for the substrate. 

The electrochemical behavior and anticorrosion performance of the composite coatings were further investigated by EIS. Figure 3a displays the Nyquist plot of all coated samples immersed in 3.5 wt.% NaCl solution. The magnitude of the Nyquist arc is closely related to the corrosion resistance of the coating. The sizes of the Nyquist arcs vary so dramatically between samples that we cannot see the smaller curves in the diagram. To clearly show the smaller ones, we display successively zoomed-in plots in Figure 3b,c. Evidently the composite coating filled with GN_0.05_ outperforms the others in terms of anticorrosion by at least one order of magnitude. The corresponding Bode plot for all samples and the equivalent electric circuit for modeling are shown in Figure 4a,b, respectively. Several curves on the Bode plot clearly show two plateaus, implying that there are two time constants. The fitted equivalent circuit parameters, including the solution resistance (*R_s_*), pore resistance (*R_p_*) and charge transfer resistance (*R_ct_*), were extracted by the ZView^®^ software Version 3.5a (Scribner Associates, Southern Pines, NC) and are summarized in Table 2. The magnitude of impedance at the low-frequency limit (|*Z*|_0.01Hz_) is regarded as a strong indicator for the barrier capability of the coating [7,10,12,25,27,41]. As can be seen from Bode plots, the |*Z*|_0.01Hz_ value for the GN_0.05_ composite by about two orders of magnitude higher than that of pure epoxy. Even though when the loading content of GN increases to 0.1 wt.% or decreases to 0.01 wt.%, |*Z*|_0.01Hz_ drops noticeably, it remains much higher than pure epoxy. Interestingly, GN_0.5_ performs even worse than pure epoxy, revealing the harmful effect of filler overload. At the optimum loading content of 0.05 wt.%, |*Z*|_0.01Hz_ is as high as 1.2 × 10^8^ Ω·cm^2^ (the corresponding value of pure epoxy coating is 1.2 × 10^6^ Ω·cm^2^). All of these data acquired from the EIS measurements are in good agreement with the polarization curve analysis. 

Finally, all samples underwent 500 h of SST. The digital photographs in Figure 5 clearly show that only the sample coated with GN_0.05_ composite stayed intact after 500 h. There is no rust or any other sign of corrosion. In contrast, rust stains appeared as early as 7 h (not shown) for the pure epoxy sample and are seen on all other samples after 100 h of SST. The GN_0.01_ coating showed suppressed corrosion as evidenced by the light brownish stain, which did not appear until after 100 h. For all samples (except GN_0.05_), numerous blisters resulting from the enrichment of water, oxygen and corrosion ions at the coating-substrate interface were found as the corrosion process proceeded. SST clearly demonstrated the superior anticorrosion ability of the GN_0.05_ composite coating, supporting potentiodynamic polarization curves and EIS tests.

It should be stated that Intergard 263 is not just an epoxy resin and the corresponding hardener FAA262 not just an amine, but a fully formulated marine coating. They are already optimized for marine applications, which means the improvements found in this study are even more important.

These tests indicate that adding too much GN (e.g., 0.5 wt.%) into the coating has a harmful effect. While it is clear that too much filler in the coating will lead to poor dispersion due to the agglomeration of fillers (the difficulty of dispersing GN uniformly in the matrix increases with increasing surface area). Such agglomeration would then deteriorate the anticorrosive performance. However, the 0.5 wt.% used in this experiment does not seem to be very high. We suspect that the reason for the loss of anticorrosion ability at the relatively low filler content in our study is the high BET specific surface area (520 m^2^/g, from the nitrogen adsorption isotherm) of our GN, which makes dispersion difficult. Thus, in accordance with this argument, the optimized filler content at 0.05 wt.% is very low compared to other studies [6,7,8,9,10,11,12,22,23,24,25,26,27,28]. The higher specific surface area corresponds to a higher degree of exfoliation, which is essential for taking full advantage of the excellent properties of graphene at a low filler content. For example, GN products with a wide range of surface area values showed significantly different microwave absorbing properties, and only the GN with the largest specific surface area can offer outstanding performance at a very low loading content [37]. 

Figure 6 compares SEM images of the surface morphology on mild steel of the spray-coated epoxy coatings (a) without and (b) with GN fillers. The coating with GN fillers (Figure 6b) has a smoother surface than the pure epoxy coating (Figure 6a). Furthermore, higher-magnification images (Figure 6c,d) indicate that while the pure epoxy coating is full of micrometer-sized pores, such pores are rare in the GN filled coating. These micro-pores were inevitably created during the curing process of epoxy resin as a result of solvent evaporation. These pores may serve as the channels for corrosive ions to penetrate through the coating, thereby weakening the protection against corrosion. Clearly, the incorporation of high-aspect-ratio GN sheets can improve the quality of epoxy resin coatings by blocking these pores. Figure 7 compares the water contact angle measurement results for the (a) pure epoxy resin and (b) GN_0.05_ composite coatings. Adding GN fillers increases the contact angle of epoxy resin from around 85° (hydrophilic) to ~101°(hydrophobic). This is advantageous for resistance against the corrosive species in aqueous solutions. Summarizing, two mechanisms are responsible for the improvement of anticorrosion performance in the optimized GN-filled epoxy coating: (1) the hydrophobicity of GN and its outstanding barrier ability against the permeation of all gases and water obstruct the diffusion of corrosive agents; (2) the 2-D geometry and high aspect ratio of GN sheets not only lengthen the diffusion paths significantly but also help block/reduce micro-pores in cured epoxy resin coatings. 

While GO can be reduced to GN at a temperature as low as 200 to 300 °C, the amount of residual oxygen functional groups depends strongly on the reduction temperature: the higher the temperature, the fewer the residual oxygen functional groups. In order to investigate the influence of the reduction temperature on GN’s anti-corrosion performance, we prepared three GN samples reduced at 300, 700 and 1100 °C, which are denoted as GN300, GN700 and GN1100, respectively. The remaining oxygen contents (determined by EA) for these different samples are shown in Table 3. The oxygen content in GO was 26.4 at.% after the Staudenmaier oxidation procedure. After the reduction at 300 °C, it was still as high as 13.2 at.%, but gradually decreased with increasing temperature and eventually dropped to merely 5.6 at.% at 1100 °C. 

The morphology of these GN samples is displayed in the SEM images in Figure 8. Reduction temperatures produced little difference in the surface morphology of GN. This indicates that exfoliation primarily occurs in the initial heating stage, and the subsequent reduction from 300 to 1100 °C is less violent, without much change in morphology. Nevertheless, milder reduction of oxygen functional groups can still take away carbon atoms (during the formation of CO_2_) and create defects on the 2-D lattice. As shown in Figure 9, the Raman spectra for GN reduced at various temperatures indicates that the intensity of D-band at ~1335/cm increases monotonously with the reduction temperature. The D-band is related to disordered carbon and defects, whereas the G-band at 1585/cm corresponds to the stretching of the ordered sp^2^ carbon atoms. The increasing *I*_D_/*I*_G_ ratio (indicator for the quality of graphene) in Figure 9 confirms that the structure of GN is more severely damaged at higher reduction temperature.

The anticorrosion capabilities of these GN fillers reduced at different temperatures were compared using EIS. The Nyquist and Bode plots of all three samples immersed in 3.5 wt.% NaCl solution are displayed in Figure 10 and Figure 11, respectively, and the fitted equivalent circuit parameters are listed in Table 4. The low-frequency-limit impedance value (|*Z*|_0.01Hz_) on the Bode plot (Figure 11), an indicator for the anticorrosion performance, is roughly equal to the size of the Nyquist arc in Figure 10. At the lowest reduction temperature, 300 °C, |*Z*|_0.01Hz_ was only 3.1 × 10^7^ Ω·cm^2^. As the temperature increased to 700 and then 1100 °C, |*Z*|_0.01Hz_ increased monotonously to 9.8 × 10^7^ Ω·cm^2^ and then 1.2 × 10^8^ Ω·cm^2^. Evidently, a higher reduction temperature helps improve the corrosion resistance of GN. Since GN1100 is of similar morphology to, and of even more lattice damage than, the other two, its advantage must come from its lowest oxygen content. It is well known that the oxygen-containing functional groups on the basal planes and edges of GO, such as hydroxyl, epoxide, carbonyl and carboxyl, make GO strongly hydrophilic and promote homogeneous dispersion of GO in water. Consequently, the residual oxygen functional groups in the GN samples reduced at lower temperatures might result in easier diffusion of H_2_O molecules between the fillers and the matrix, facilitating the transport of corrosive agents.

The results of 500-h SST for the three coatings are displayed in Figure 12. Even though all samples are rust-free, blistering does occur in the GN300-filled coating, indicating corrosion at the coating-substrate interface. The SST results are in good agreement with the EIS tests.

### 3.2. Anti-Corrosion and IR-Camouflage Performances of Composite Coatings with Al-GN Fillers

To reduce the IR-emissivity of the epoxy-GN composite coatings, Al powder was added to the composite. The XRD profile and SEM image for the Al powder are shown in Figure 13a,b, respectively. The typical (111), (200), (220), (311) and (222) diffraction peaks for a face-centered cubic Al crystal are clearly observed in Figure 13a, indicating good crystallinity. The SEM image in Figure 13b reveals two types of particles of distinct shapes: one is larger and flake-like; the other is smaller and ball-like.

EIS measurements were next carried out for composite coatings filled with 0.05 wt.% of GN1100 and various amounts (5, 15, 25 and 35 wt.%) of Al. Figure 14 and Figure 15 display the Nyquist and Bode plots, respectively, for these composite coatings. Table 5 summarizes the fitted equivalent circuit parameters. The sizes of the Nyquist arcs for all samples are clearly much larger than that of GN_0.05_-epoxy. Furthermore, |*Z*|_0.01Hz_ increased slightly as Al filler was added into the composite. These results indicate that introducing Al in the coating improves rather than weakens the anticorrosion ability. This is a desirable result because we do not want to gain IR stealth at the expense of anticorrosion performance. The mechanism for the improved anticorrosion by Al powder is depicted in Figure 16. The Al flakes, having a planar geometry similar to GN but with a much larger lateral size, serve as barriers for the diffusion of water and Cl^−^ and make the pathway of the aggressive agents even more tortuous.

To verify the anticorrosion capability of the composite coating, the 500-h SST was also carried out. The visual appearances of the coating with Al_25_-GN_0.05_ fillers before and after SST are shown in Figure 17. The corresponding photographs for the GN-epoxy coating with the optimized condition are also displayed for comparison. Both coatings stayed intact after the test with no signs of rust or blistering, confirming their excellent corrosion resistance.

Kirchhoff’s law of thermal radiation states that for an arbitrary body emitting and absorbing thermal radiation in thermal equilibrium, the emissivity (*ε*) and absorptivity (*α*) must be equal. Thus, a good absorber must be a good emitter. Moreover, for opaque objects, *α* = 1 − *R*, where *R* is the reflectivity. Therefore, a good reflector must be a poor absorber/emitter. The high absorptivity of GN fillers inevitably increases the emissivity of the coating. To counteract this harmful effect, we added highly reflective Al pigment in the composite. Figure 18 displays the IR images (with thermal emissivity values) for the pure epoxy, GN_0.05_-epoxy and Al_25_-GN_0.05_-epoxy composites. Note that emissivity is a dimensionless number between 0 (for a perfect reflector) and 1 (for a perfect absorber/emitter, i.e., a blackbody). The average emissivity of neat epoxy resin was 0.75 in the 3–5 μm window and 0.93 in the 8–12 μm window. These numbers indicate that the epoxy coating absorbs IR very well, and thus has a high emissivity. Note that emissivity *ε* is the efficiency that a material radiates energy compared to a blackbody. So *ε* = 1 for a blackbody. If an object has a constant *ε* (<1), i.e., *ε* does not depend on wavelength, then it is called a graybody. Most of the real-world objects belong to the third category—selective radiator: their emissivity varies both with wavelength and temperature, i.e., *ε* = *f*(*λ*, *T*). So the emissivity values in 3–5 μm and 8–12 μm windows might be quite different in a material. The value of *ε* also depends on other factors such as surface treatment and material thickness. Adding 0.05 wt.% of GN (1100 °C) in the epoxy raised the IR emissivity to 0.98 and 0.94 in 3–5 μm and 8–12 μm, respectively. These increases are due to the high absorption of GN in the IR wavelengths, which is detrimental in the aspect of IR camouflage. After introducing low-thermal-emissivity aluminum powder (25 wt.%) to address this issue, the average emissivity values in the two windows dropped to 0.63 and 0.64, respectively, which were even lower than the blank epoxy coating (by 16% and 31%). 

Figure 19 compares the average thermal emissivity values in the 3–5 and 8–12 μm windows for the neat epoxy, GN_0.05_-epoxy and Al-GN_0.05_-epoxy coatings with various Al contents (5, 15, 25 and 35 wt.%). The effect of the Al content on the IR emissivity in both windows are clearly seen, adding 15–35 wt.% of Al helped to reduce the thermal emissivity significantly. Even at 5 wt.% of Al content, the harmful impact of GN on IR emissivity can be greatly alleviated. The results indicate that Al-GN-epoxy composite coatings can not only protect against corrosion but also have good potential for the applications requiring IR camouflage.

## 4. Conclusions

In this study, the corrosion resistance and IR emissivity of GN-epoxy and Al-GN-epoxy coatings were studied. The effects of GN concentration and GO thermal reduction temperatures along with Al concentration were characterized. Polarization curves, EIS and SST confirmed that the anticorrosion performance of epoxy was effectively improved by adding GN and was optimized at an ultralow GN content of 0.05 wt.%, (advantageous for cost minimization). Higher thermal reduction temperatures of GN improved anticorrosion characteristics significantly (e.g., 290% increase in |*Z*|_0.01Hz_ when the reduction temperature was raised from 300 °C to 1100 °C) due to the reduction of residual oxygen functional groups. A |*Z*|_0.01Hz_ as high as 1.2 × 10^8^ Ω·cm^2^ was achieved for GN reduced at 1100 °C. In the aspect of IR stealth, the epoxy itself exhibited high IR emissivity, which is undesirable, and adding the GN filler increased the IR emissivity even more. Fortunately, introducing 15–35 wt.% of Al powder into the composite coating not only effectively reduced the IR emissivity (which can be even lower than that of blank epoxy coating) but also retained the good anticorrosion capability. The Al_25_-GN_0.05_-epoxy coating successfully protected the steel substrate throughout the 500 h of SST without any sign of rust or blistering. We believe that Al-GN-epoxy composites will be useful for the application of IR-stealth anticorrosion coatings. 

## Figures and Tables

**Figure 1 nanomaterials-11-01603-f001:**
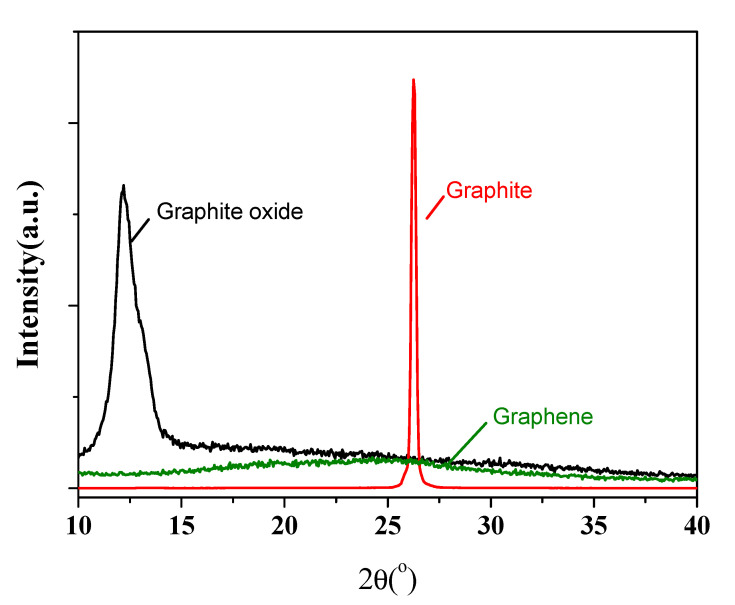
XRD results for natural graphite, Staudenmaier GO and graphene.

**Figure 2 nanomaterials-11-01603-f002:**
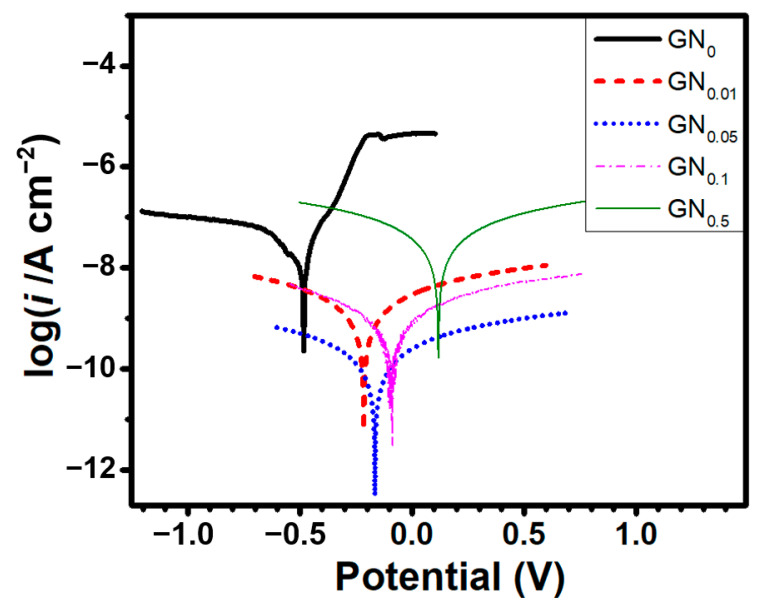
Potentiodynamic polarization curves for samples coated with epoxy added with various amounts (0, 0.01, 0.05, 0.1 and 0.5 wt.%) of GN.

**Figure 3 nanomaterials-11-01603-f003:**
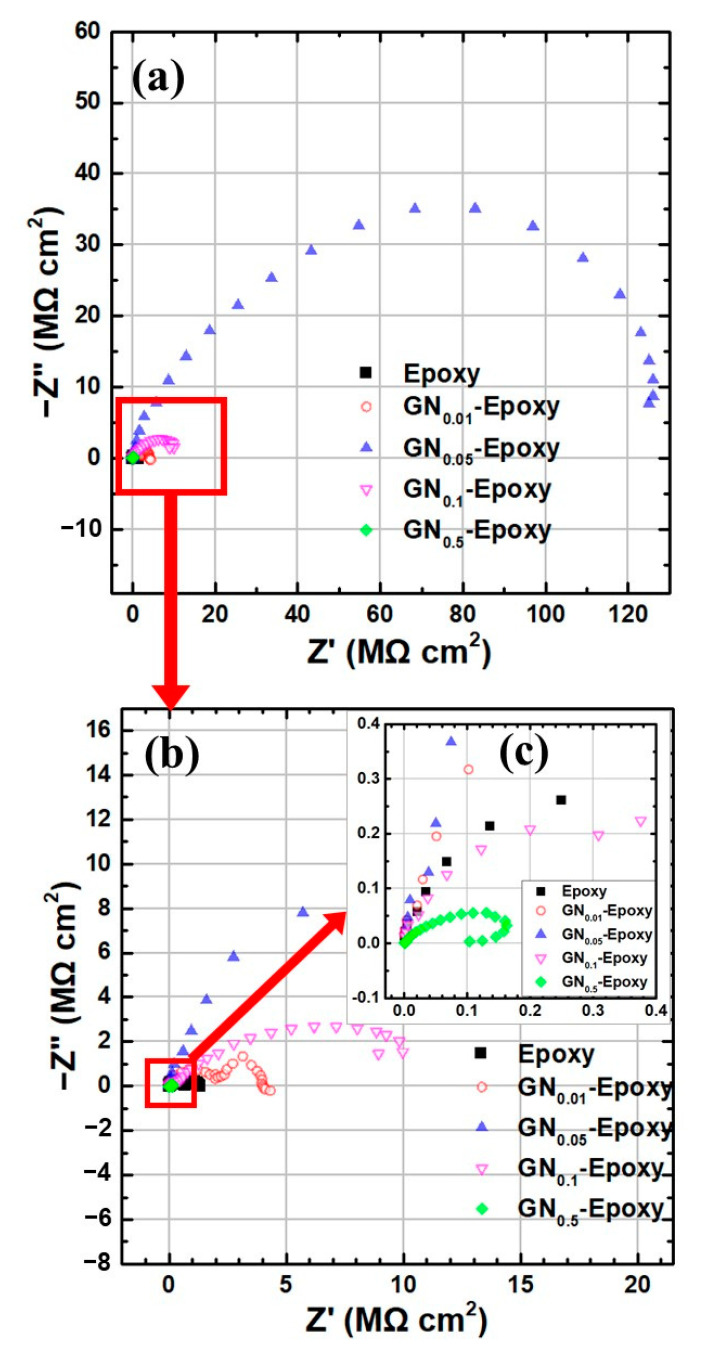
Nyquist plots of EIS for samples immersed in 3.5 wt.% NaCl solution. (**b**,**c**) are the successively magnified views of (**a**).

**Figure 4 nanomaterials-11-01603-f004:**
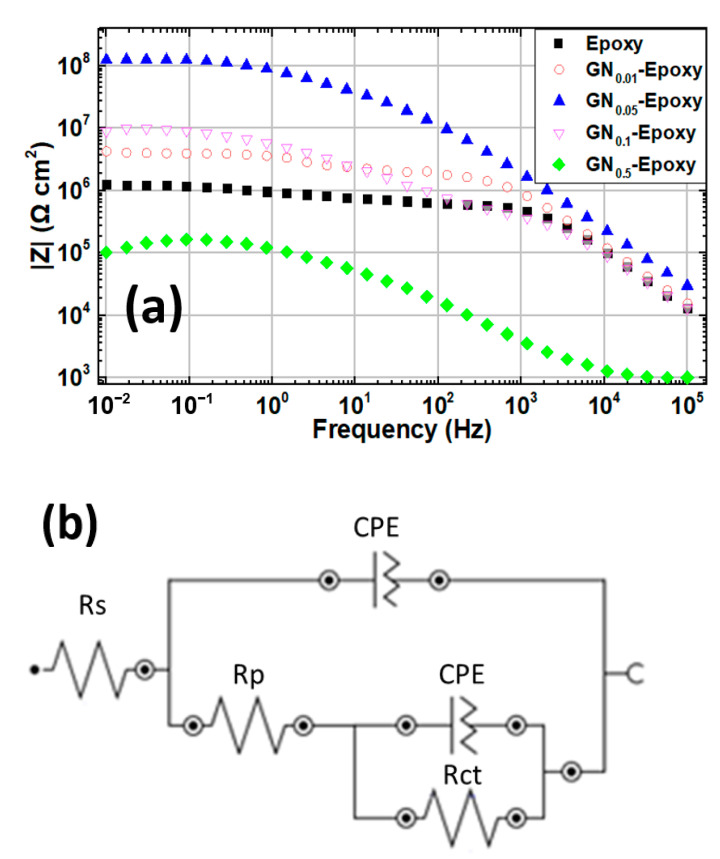
(**a**) Bode plots of EIS for samples immersed in 3.5 wt.% NaCl solution. (**b**) Equivalent electric circuit for modeling.

**Figure 5 nanomaterials-11-01603-f005:**
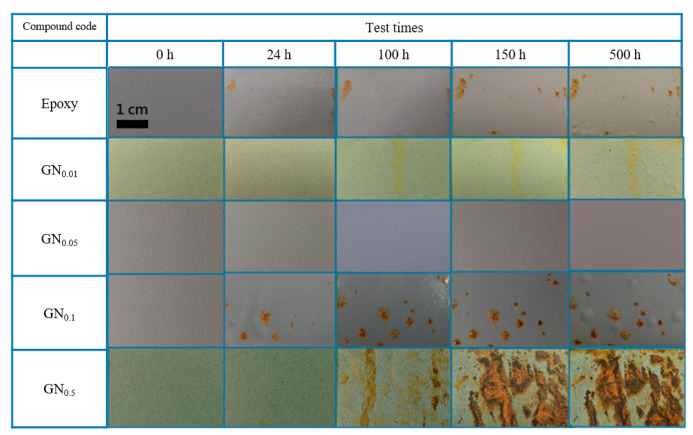
Visual appearance of samples (0.00, 0.01, 0.05, 0.1 and 0.5 wt.% of GN) following 0, 24, 100, 150 and 500 h of SST.

**Figure 6 nanomaterials-11-01603-f006:**
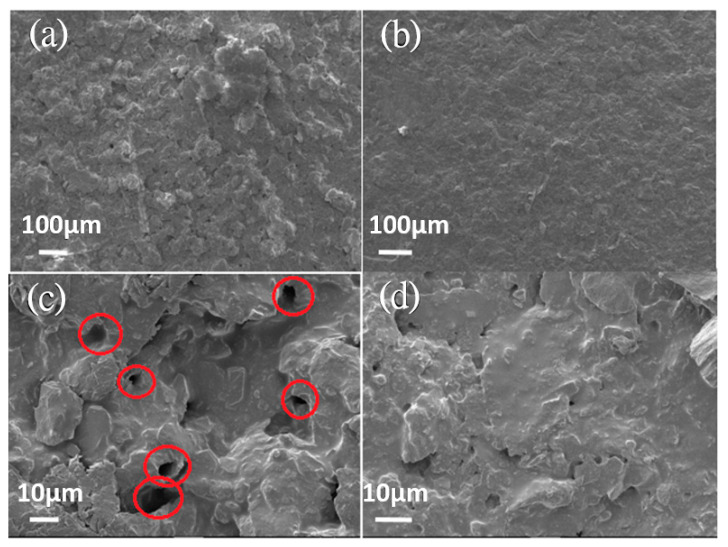
SEM images of the surface morphology for the coatings (**a**) without and (**b**) with GN fillers. (**c**,**d**) are the high-magnification images for (**a**,**b**), respectively.

**Figure 7 nanomaterials-11-01603-f007:**
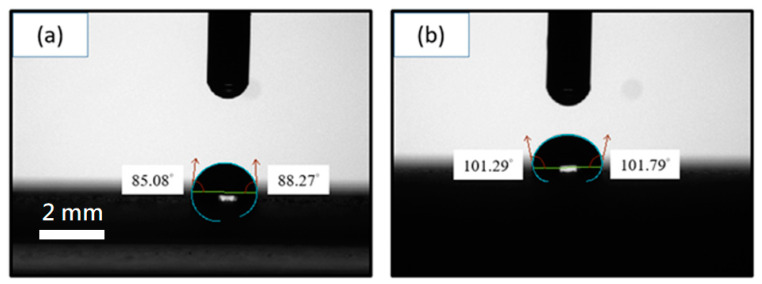
Water contact angle measurements for the epoxy resin coatings: (**a**) without GN; (**b**) with 0.05 wt.% of GN. (The scale bars for (**a**) and (**b**) are the same.).

**Figure 8 nanomaterials-11-01603-f008:**
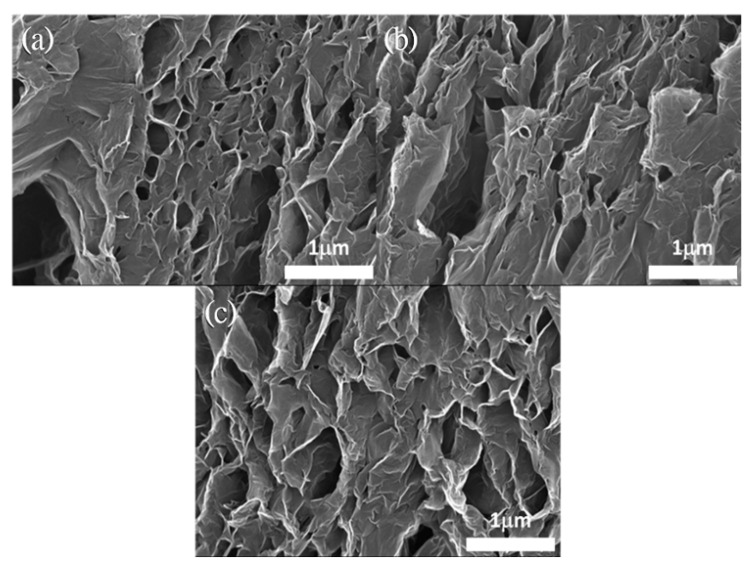
SEM morphology of (**a**) GN300, (**b**) GN700 and (**c**) GN1100.

**Figure 9 nanomaterials-11-01603-f009:**
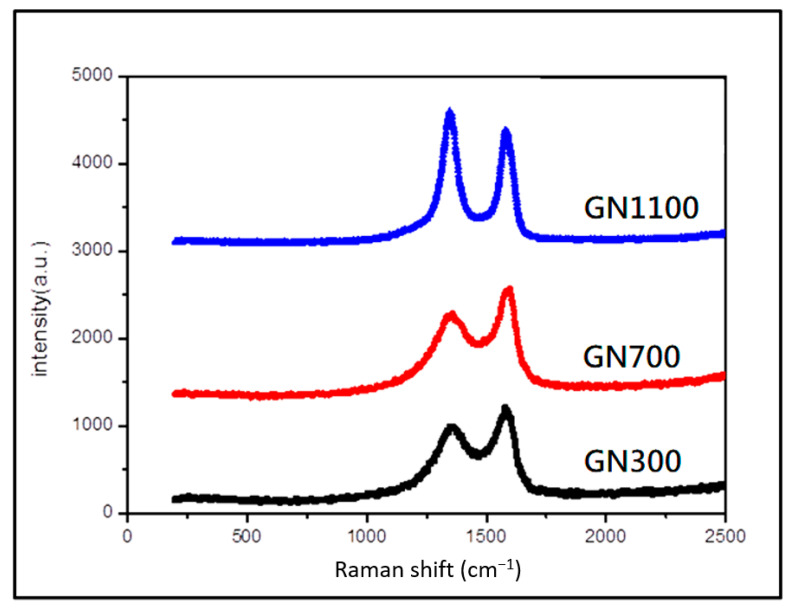
Raman spectra for GN reduced at various temperatures.

**Figure 10 nanomaterials-11-01603-f010:**
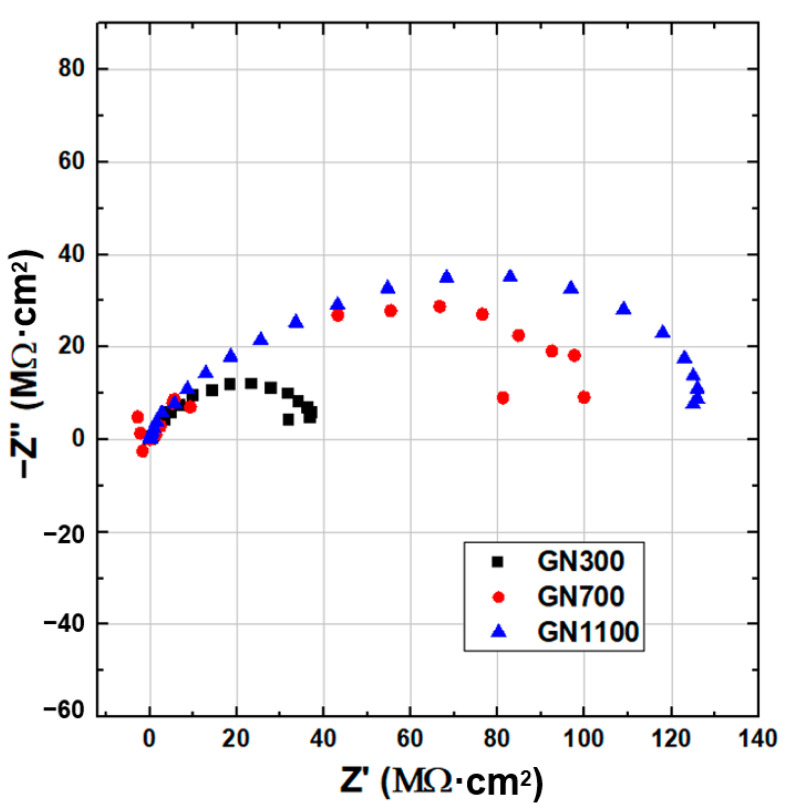
Nyquist plots of EIS for coatings filled with 0.05 wt.% of GN reduced at different temperatures (GN300, GN700 and GN1100) immersed in 3.5 wt.% NaCl solution.

**Figure 11 nanomaterials-11-01603-f011:**
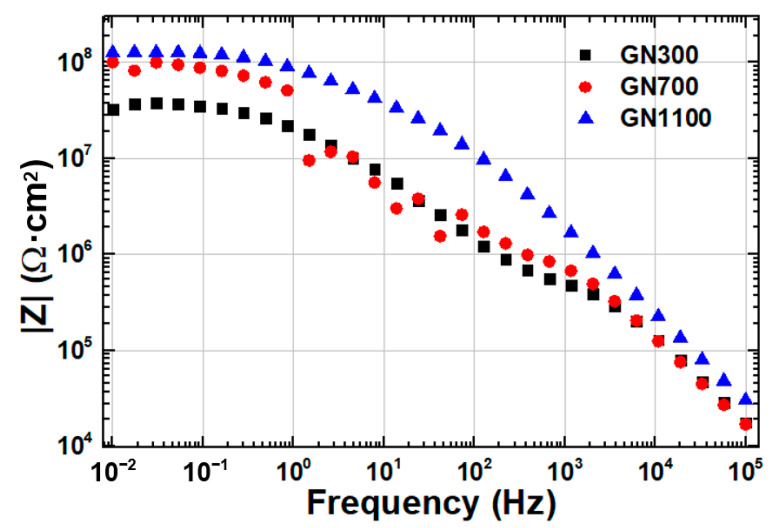
Bode plots of EIS for coatings filled with 0.05 wt.% of GN300, GN700 and GN1100 immersed in 3.5 wt.% NaCl solution.

**Figure 12 nanomaterials-11-01603-f012:**
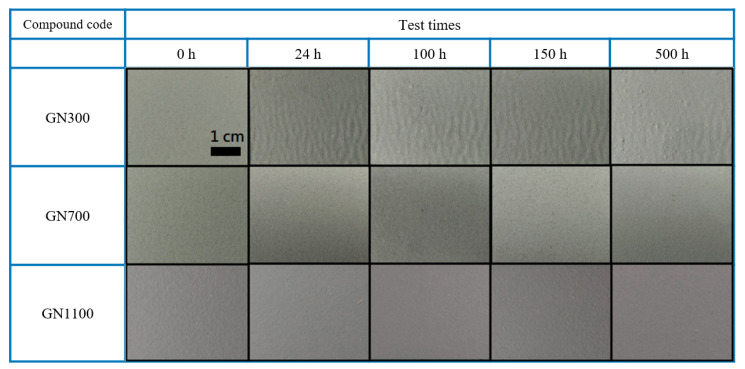
Visual appearances of all samples after various durations (0, 24, 100, 150 and 500 h) of SST.

**Figure 13 nanomaterials-11-01603-f013:**
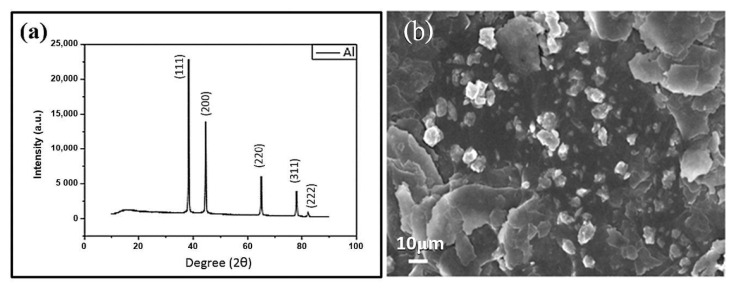
(**a**) XRD and (**b**) SEM image for the Al powder.

**Figure 14 nanomaterials-11-01603-f014:**
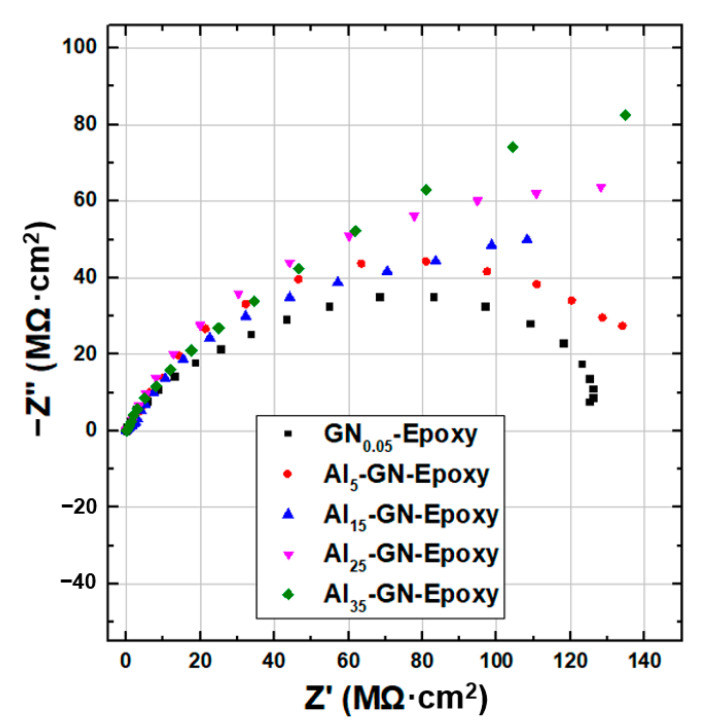
Nyquist plots of EIS for coatings filled with 0.05 wt.% of GN1100 and various contents (5, 15, 25 and 35 wt.%) of Al immersed in 3.5 wt.% NaCl solution.

**Figure 15 nanomaterials-11-01603-f015:**
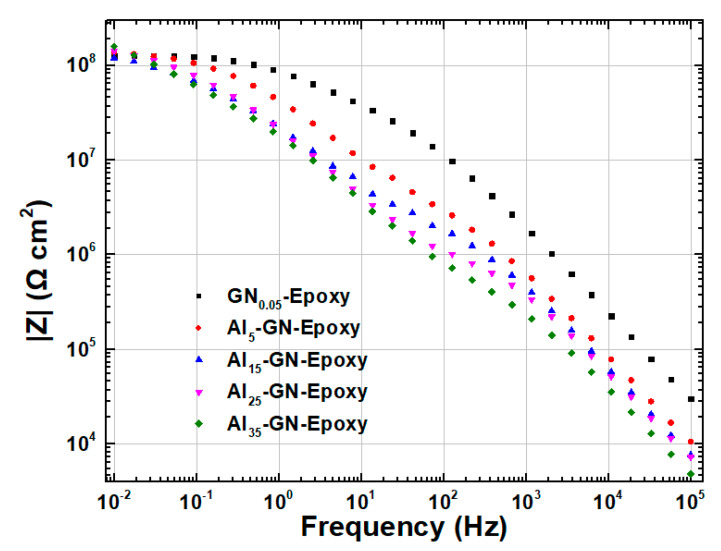
Bode plots of EIS for coatings filled with 0.05 wt.% of GN1100 and various contents (5, 15, 25 and 35 wt.%) of Al immersed in 3.5 wt.% NaCl solution.

**Figure 16 nanomaterials-11-01603-f016:**
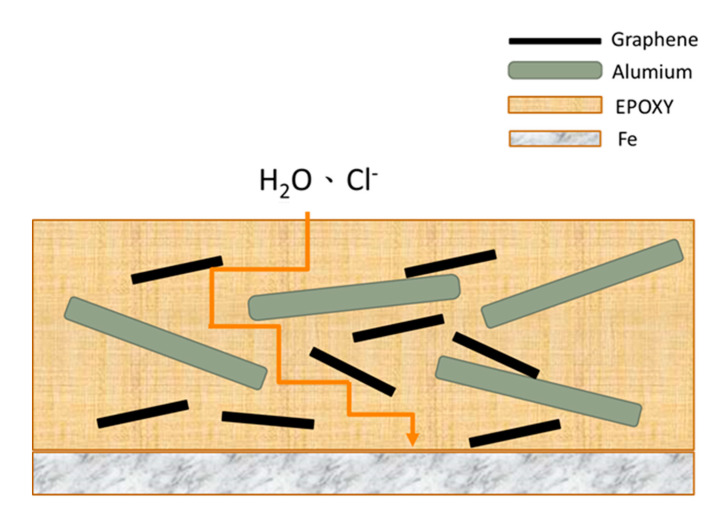
Schematic for the mechanism of improved anticorrosion in epoxy coatings with Al-GN fillers.

**Figure 17 nanomaterials-11-01603-f017:**
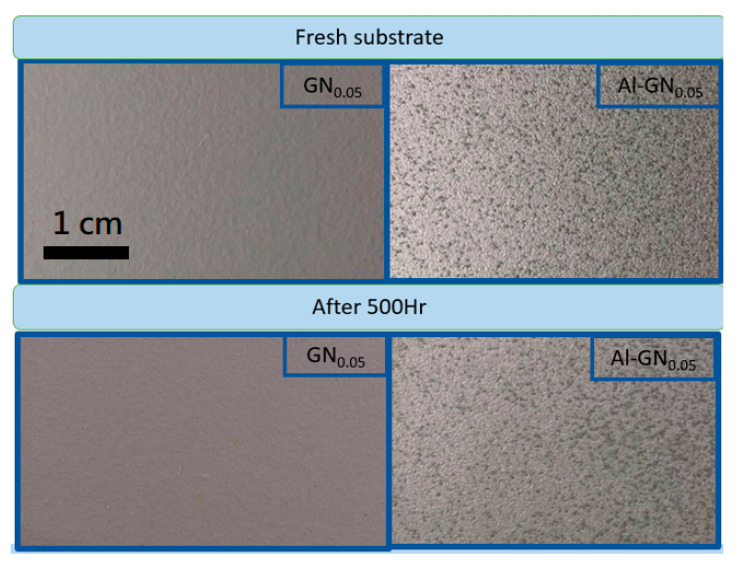
Visual appearances of the coatings with GN_0.05_ and Al_25_-GN_0.05_ fillers before and after SST.

**Figure 18 nanomaterials-11-01603-f018:**
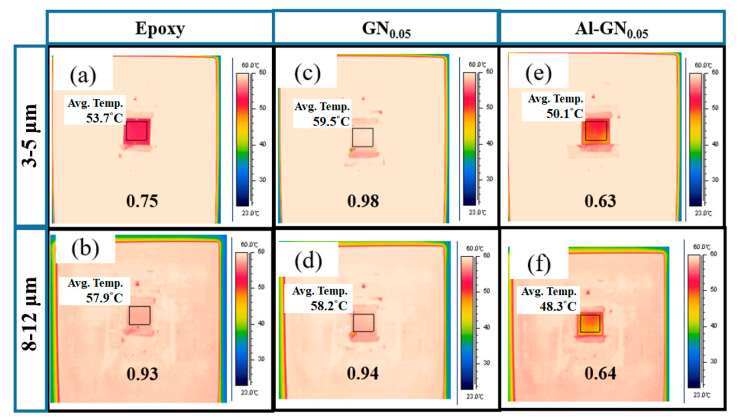
IR images (with average thermal emissivity values and average temperatures) in the 3–5 μm and 8–12 μm windows for (**a**,**b**) the neat epoxy, (**c**,**d**) GN_0.05_-epoxy and (**e**,**f**) Al_25_-GN_0.05_-epoxy coatings.

**Figure 19 nanomaterials-11-01603-f019:**
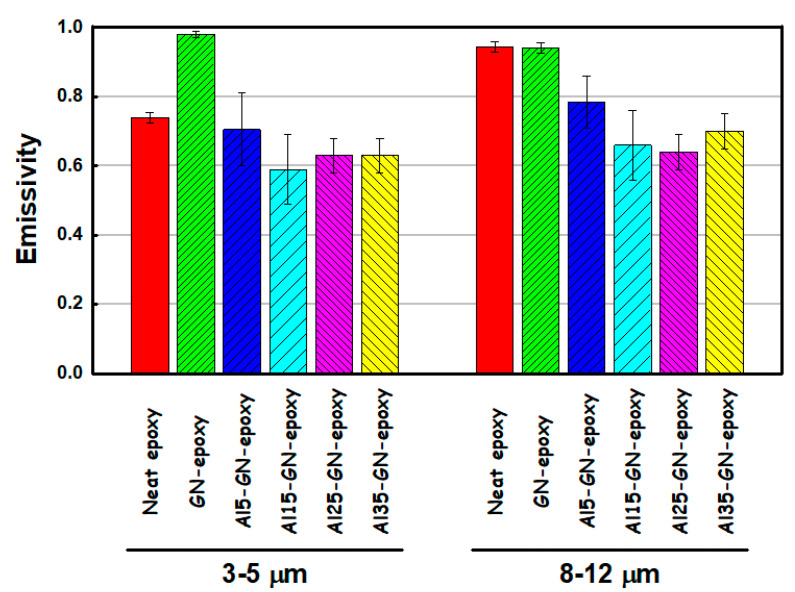
Average thermal emissivity values in the 3–5 and 8–12 μm windows for the neat epoxy, GN_0.05_-epoxy and Al-GN_0.05_-epoxy coatings with various Al contents (5, 15, 25 and 35 wt.%).

**Table 1 nanomaterials-11-01603-t001:** Electrochemical corrosion parameters for bare substrate, epoxy- and GN-epoxy coated samples immersed in 3.5 wt.% NaCl solution.

Sample	*E_corr_* (V)	*i_corr_* (A/cm^2^)	*β_c_* (V/Dec)	*β_a_* (V/Dec)	*v_corr_* (mm/Year)
GN_0_	−0.48	1.02 × 10^−7^	−0.20	0.096	1.19 × 10^−3^
GN_0.01_	−0.21	3.89 × 10^−10^	−0.121	0.117	4.52 × 10^−6^
GN_0.05_	−0.16	4.57 × 10^−11^	−0.114	0.115	5.31 × 10^−7^
GN_0.1_	−0.09	2.69 × 10^−10^	−0.111	0.112	3.13 × 10^−6^
GN_0.5_	0.12	8.12 × 10^−9^	−0.115	0.117	9.44 × 10^−5^

**Table 2 nanomaterials-11-01603-t002:** Fitted equivalent circuit parameters of EIS for samples immersed in 3.5 wt.% NaCl solution.

Sample	*R_s_* (Ω·cm^2^)	*R_p_* (Ω·cm^2^)	*R_ct_* (Ω·cm^2^)
epoxy	0.01	5.2 × 10^5^	8.6 × 10^5^
GN_0.01_	0.01	1.9 × 10^6^	2.3 × 10^6^
GN_0.05_	0.01	2.5 × 10^7^	1.1 × 10^8^
GN_0.1_	0.01	1.1 × 10^7^	1.3 × 10^7^
GN_0.5_	0.01	3.6 × 10^3^	2.3 × 10^5^

**Table 3 nanomaterials-11-01603-t003:** Oxygen contents for GO and different GN samples.

Reduction Temperature	Sample Weight (mg)	O (at.%)
GO (not reduced)	2.5	26.4
300 °C	2.5	13.2
700 °C	2.5	10.0
1100 °C	2.4	5.6

**Table 4 nanomaterials-11-01603-t004:** Fitted equivalent circuit parameters of EIS for coatings filled with 0.05 wt.% of GN300, GN700 and GN1100 immersed in 3.5 wt.% NaCl solution.

Sample	*R_s_* (Ω·cm^2^)	*R_p_* (Ω·cm^2^)	*R_ct_* (Ω·cm^2^)
GN300-epoxy	0.01	3.8 × 10^5^	4.2 × 10^7^
GN700-epoxy	0.01	1.3 × 10^6^	9.0 × 10^7^
GN1100-epoxy	0.01	2.5 × 10^7^	1.1 × 10^8^

**Table 5 nanomaterials-11-01603-t005:** Fitted equivalent circuit parameters of EIS for coatings filled with 0.05 wt.% of GN1100 and various contents (5, 15, 25 and 35 wt.%) of Al immersed in 3.5 wt.% NaCl solution.

Sample	*R_s_* (Ω·cm^2^)	*R_p_* (Ω·cm^2^)	*R_ct_* (Ω·cm^2^)
GN_0.05_-epoxy	0.01	2.47 × 10^7^	1.14 × 10^8^
Al_5_-GN_0.05_-epoxy	0.01	4.05 × 10^6^	1.48 × 10^8^
Al_15_-GN_0.05_-epoxy	0.01	3.54 × 10^7^	1.60 × 10^8^
Al_25_-GN_0.05_-epoxy	0.01	1.01 × 10^6^	1.67 × 10^8^
Al_35_-GN_0.05_-epoxy	0.01	1.27 × 10^6^	2.49 × 10^8^

## Data Availability

The data presented in this study are available on request from the corresponding authors.

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
