# Peer review of "Using Graphene-Based Composite Materials to Boost Anti-Corrosion and Infrared-Stealth Performance of Epoxy Coatings"

_nanomaterials, 2021, doi:10.3390/nano11061603_

Round 1
Reviewer 1 Report
This is a very nice publication carefully put together, well written, and well explained, with lots of work and experiments having been done, and lots of useful data. A very complete manuscript. Only the part “3.2 Anti-corrosion and IR-camouflage performances of composite coatings with Al-GN fillers” is a little bit confusing since it is not clear. The following should also be considered:
- The methodology section is not well organized for the readers to understand the concept.
- The result of “2. potentiodynamic polarization curves” is superficial. The authors should try to provide more explanations.
- There are many linguistic flaws in the article. Check the article for writing and language.
- Figs. 5,7,12, and 17 need to scale bar.
Reviewer 2 Report
Suggestion1)
Abstract (page 2, line 30) instead of "...thermally reduced.." use "...which have been thermally reduced..." to clarify the graphene manufacturing process is meant.
Suggestion 2)
It should be stated that Intergard 263 is not an epoxy resin and the corresponding hardener FAA262 not just an amine, but a fully formulated marine coating. Intergard is already optimized for marine applications, which means the improvements found are even more important... Furthermore it should be mentioned that Intergard 263 cotains 25 wt% of aromatic solvents and approx. 5 wt% of benzyl alcohol which certainly will interact with graphene platelets - effects unknown. It would be extremely helpful if a standard DGEBA epoxy resin (i.e. Epikote 828 from Hexion), cured with the stochiometric amount of of FAA262 would be investigated as well to estimate the influence of the solvents (eventually improved/easier dispersion of graphene in the coating?!).
Reviewer 3 Report
This manuscript describes a study about the incorporation of graphene nanosheets to epoxy coating as barrier coating against corrosion, in different proportion and different conditions for its reduction. Another study about the addition of Al powder to that composite in order to reduce the IR emissivity. In general this work is well planned, well described and well developed and the author´s contributions are innovative. However, there are some aspects that should be improved or clarified to increase the quality of the manuscript. They are the following:
- Lines 97-101, insert references or describe the procedures for dispersion, homogenization, mixing, deposition to get the appropriate thickness and measuring coating thickness. They are poorly described.
- 2 needs to improve quality.
- Symbol forms should be changed (triangle, square…) in Fig.2, 3, 4, 9, 10 , 11, 14, 15 in order to facilitate the reading and better distinguish the different options in printed form in black&white.
- Coating thickness columns should be removed from the tables 1, 4 and 5. They do not give any information.
- Table 1, title Icorr should be changed by icorr in accordance to the text.
- Table 3 title, please put GN instead GE.
- 6, 8, 13b Put (a), (b), (c), (d) and references in µm in white font because white letters stand out clearly against a black background.
- Line 312, a mistake in the first word: increased
- Axis title and number sizes should be increase for an easier reading in the graphics.
- The use of Al 25 wt.% is the best option? It should be explained anywhere.
- Fig 18. 3-5 µm and 8-12 µm should be rotated 90º in the figure. Average temperatures are in small letters, please increase the size. Could the authors explain the reason for the big between emissivity values for epoxy coating?
- Fig 18 and 19. Please insert the units for emissivity.
- Why the fig. 19 is not complete? Could the authors include the bars for all the Al contents? And put the bars in different frame to distinguish them in black and white print. Perhaps including the percentages in the x-axis it could facilitate the understanding.
- Conclusions should be more categorical. For example, in line 369 affected, but how?, or in line 372, Al powder but what amount?
Round 2
Reviewer 1 Report
The revised manuscript is improved. However, the result of “2. potentiodynamic polarization curves” is superficial, yet. The authors should try to provide more and deep explanations and details.
